# Experiences of Veterans, Caregivers, and VA Home-Based Care Providers before, during, and Post-Hurricane Ian

**DOI:** 10.3390/geriatrics9010010

**Published:** 2024-01-10

**Authors:** Leah M. Haverhals, Chelsea Manheim, Deisy Vega Lujan

**Affiliations:** 1Denver-Seattle Department of Veterans Affairs (VA) Center of Innovation for Veteran-Centered & Value-Driven Care, Rocky Mountain Regional VA Medical Center, Aurora, CO 80045, USA; chelsea.manheim@va.gov (C.M.); deisy.vega-lujan@va.gov (D.V.L.); 2Health Care Policy & Research, School of Medicine, University of Colorado Anschutz Medical Campus, Aurora, CO 80045, USA

**Keywords:** in-home care, caregivers, long-term care, Veterans, Hurricane Ian

## Abstract

On 28 September 2022, Hurricane Ian pummeled parts of south Florida. Disaster and climate change research has shown that disasters exacerbate inequalities, especially amongst older and physically vulnerable people. Florida has a large population of Veterans managing multiple chronic health conditions and receiving long-term care in-home from Veterans Health Administration (VA) programs, including Home Based Primary Care and Medical Foster Home. To describe how VA staff provided high quality care during and after the hurricane, and how Veterans and caregivers accessed needed healthcare and supports post-hurricane, we conducted a site visit to Lee County, Florida area in May 2023, conducting N = 25 interviews with VA staff, Veterans, and caregivers. Findings from qualitative thematic analysis showed that while some Veterans and caregivers experienced significant challenges during and after the hurricane, including displacement and difficulty accessing oxygen, they felt highly supported by VA care teams. Staff efforts post-hurricane focused on improving care coordination in anticipation of future disasters, especially around communicating with Veterans and their caregivers, and a VA workgroup formed to implement changes. As climate change causes more severe hazard events, lessons learned from this project can better support healthcare staff, older adults, and their caregivers before and after major disasters.

## 1. Introduction

On 28 September 2022, Hurricane Ian made landfall as a Category 4 hurricane, pummeling areas of southwest Florida. Ian caused extensive flooding, damage to physical and cellular infrastructure, loss of power, displaced people from their homes, and caused considerable deaths [1]. It has been ranked the fifth strongest hurricane to hit the contiguous United States (US), with federal support totaling $8.7 billion [2,3]. Disaster and climate change research has shown that disasters exacerbate inequalities, especially amongst older and physically vulnerable people [4,5]. A recent systematic review of 120 publications in the areas of disaster management, urban areas, and gerontology found that older adults’ needs were not being met during the critical period of disaster response [6]. Florida has a large population of both older adults [7] and older Veterans [8] managing multiple chronic health conditions and receiving long-term services and supports in-home. Many of these individuals are primarily homebound and often live alone. A recent study of older adults in the Miami-Dade County, Florida area used geospatial analysis to determine where adults older than 65 lived in relation to flood-prone areas, finding that those living alone and in older homes were especially vulnerable to managing the impacts of flooding [9]. Studies have also shown that older adults who are homebound experience more risk factors for loneliness than those who are not homebound [10]. These include more difficulty with activities of daily living, loss of relationships, cognitive impairment, depression, lack of social support, and limited mobility [11,12,13]. Older adults managing these factors become increasingly vulnerable during and after major disasters like Hurricane Ian [14].

For Veterans who have highly complex medical conditions and are nearing nursing home eligibility, the US Department of Veterans Affairs (VA) offers several home and community-based service options to best support these types of Veterans and their caregivers. Two of these programs are VA Home Based Primary Care (HBPC) and the VA Medical Foster Home (MFH) program [15,16,17]. Both of these programs are well established, with interdisciplinary care team members in HBPC providing in-home healthcare to both HBPC and MFH Veterans. There are currently 121 MFH programs across the US [18] caring for over 6000 Veterans since the program’s inception in 1999 [19]. Both the MFH program and the HBPC programs are expanding to all VAMCs by the end of the US federal fiscal year 2026 [20]. In 2021, HBPC teams cared for nearly 60,000 Veterans across 70 VA Medical Centers (VAMCs) and 300 Community-based Outpatient Clinics (CBOCs) [21]. Veterans in HBPC reside in their own home, living alone or with family or caregiver(s), while MFH Veterans are placed in the home of a non-VA caregiver that the coordinator of the MFH program and other VA care team members recruit and screen. Past studies have shown Veterans often become like part of the MFH caregiver’s family, normally living there until the end of their life [22]. The same has been shown of HBPC Veterans [21,23]. In recent years, the VA has also piloted a program called Redefining Elder Care in America or RECAP program, which uses predictive analytics to identify Veterans in need of long-term services and supports to better support them at home and delay placement in long-term care facilities like nursing homes [24].

As the older adult population continues to grow along with their wishes to age in place in their own homes, it is important to study how older adults with complex health histories—as well as their healthcare teams and caregivers—manage before, during, and after major disasters like Hurricane Ian. Therefore, the objective of this quality improvement project was to describe how VA staff provided high-quality care during and after Hurricane Ian, how Veterans and caregivers accessed needed healthcare and supports post-hurricane, and lessons learned to better support older Veterans, their caregivers, and healthcare teams who care for older adults in the event of future disasters.

## 2. Materials and Methods

### 2.1. Participants and Recruitment

We designed a descriptive qualitative study to conduct semi-structured interviews during an in-person site visit with VA Staff from the Lee County, Florida VA Community Based Outpatient Clinic (CBOC). This included recruiting multi-disciplinary providers from HBPC, MFH, and the RECAP program. Additionally, we recruited Veterans and Veterans’ caregivers who receive care from the HBPC and MFH programs to be interviewed.

To recruit VA staff, we first contacted a member of the Lee County HBPC leadership, explained the study, and answered questions they had regarding participation. Following their agreement, they invited all members of their HBPC and MFH teams to participate in our project during a team meeting. This VA staff member provided our project team with the names of 12 staff who agreed to be interviewed. We emailed these staff and notified them of the dates of our site visit. There were some staff that were unavailable during our site visit and agreed to be interviewed over the phone. 

We requested assistance from the HBPC team, via the HBPC program director, to assist with identifying Veterans, family caregivers, and MFH caregivers who would be willing to be interviewed about their experiences around Hurricane Ian. We received a list of (N = 17): Veterans (n = 12), family caregivers (n = 1), and MFH caregivers (n = 4). We contacted Veterans and caregivers by phone to schedule interviews during the site visit. When we did not reach them by phone, we left up to two voicemails, providing them with information about the project and inviting them to participate, with the assurance that participation was voluntary and confidential. Two Veterans and one MFH caregiver requested to be interviewed by phone rather than have us visit in-person. Five Veterans declined or were unreachable, and two MFH caregivers were unreachable. We scheduled interviews at times convenient for participants.

### 2.2. Data Collection

We created semi-structured interview guides (See Appendix A) specific to each participant type. All guides were designed to gather information on participants’ experiences before, during, and after Hurricane Ian. For staff interviews, we also asked about hurricane-preparation planning at the facility and by the team. For Veteran and caregiver interviews, some questions focused on VA care teams’ roles supporting them before, during, and after, as well as if they had experienced a disaster like this previously.

Between April 2023 and May 2023, three team members conducted interviews with VA staff, Veterans, and Veterans’ caregivers in person and over Microsoft Teams. Participants included (n = 11) VA staff members (n = 7 in-person and n = 4 over Microsoft Teams), (n = 8) Veteran interviews, (n = 2) MFH caregiver interviews, and (n = 3) family caregiver interviews, for a total of (N = 24) interviews. Note that some family caregivers were referred to us after receiving the original list of caregivers interested in participating. Staff roles included social workers, nurses, dieticians, recreational therapists, physical therapists, medical doctors, and program specialists (See Figure 1 and Figure 2 for Staff Characteristics, Table 1 for Veteran Characteristics, and Table 2 for Caregiver Characteristics).

This study was deemed a quality improvement project by the Rocky Mountain Regional VA Research & Development Review Committee. We received VA national union concurrence from The Local Supplemental Agreement between the Bay Pines VA Healthcare System and the AFGE Local 548 to conduct interviews with VA staff. As this was a quality improvement project, we were not required to obtain Institutional Review Board approval nor documentation of written informed consent. Prior to each interview, we obtained verbal consent from participants to both participate in the interview and to audio record it. Interview duration ranged from 18 to 58 min, with the average length being 42 min. All interviews were transcribed verbatim by a VA-approved transcription service.

### 2.3. Data Analysis

We applied a combined inductive and deductive approach to thematic analysis of the interview data [25]. Two qualitative analysts and the team methodologist created deductive code lists independently based on questions from each interview guide and met to reach agreement on the deductive codes. We used Atlas.ti version 9.0 qualitative analytic software [26], and all three members initially coded three interviews independently, meeting subsequently to compare codes and reach consensus on code application and definitions. We divided the remaining transcripts between us, meeting weekly to discuss codes emerging from the data and early themes. We completed line-by-line coding in mid-July 2023 and continued analysis by querying data to review and develop themes describing the experiences of VA staff, Veterans, and caregivers with hurricane Ian.

## 3. Results

We identified five themes (Figure 3) reflective of VA staff providing high-quality care during and after hurricane Ian, and explaining how Veterans, caregivers, and VA staff managed post-hurricane. These included: (1) staff’s prioritization of disaster preparation and making proactive changes to ensure high-quality care in the future; (2) Veterans and caregivers feeling highly supported by VA providers; (3) Veterans and caregivers experiencing significant challenges due to the hurricane, and relying heavily on their social support networks post-hurricane; (4) VA staff managing the attitudes of Veterans not taking the hurricane seriously; and (5) VA staff, Veterans, and caregivers sharing the lessons they learned.

### 3.1. Staff’s Prioritization of Disaster Preparation and Making Proactive Changes to Ensure High-Quality Care in the Future

Being located in Florida, the VA care teams were well-versed in disaster preparation for hurricanes. Staff described calling every Veteran on their patient panel a few days before Ian hit as a best practice, as well as reviewing their emergency plans annually. One HBPC staff member noted,


*We call everyone. Make sure they have food, water, a place to go, that they have the plan… soon as they come into the program, but we kind of make sure the plan has not changed. Try to find out where they’re going. So we can do post [storm] calls if they have to leave. I’ll kind of let them know where the shelters are if they need that.*

*(Staff 109)*


Another HBPC staff member shared that when they called Veterans and caregivers two to three days before Ian, they made sure they had the following:


*Enough oxygen tanks, make sure they had enough medicine if they’re gonna be down for two weeks with no electricity. We asked if they had a generator? Did they have a place to go? Did they have enough food? Did they have enough water? All those kind of questions [are] in our template.*

*(Staff 103)*


Veterans and their caregivers reported having most or all of the medications they needed before Ian hit. Additionally, the VA allowed Veterans to obtain medications and prescriptions from non-VA pharmacies in advance of Ian, including Walgreens, CVS, and Walmart.


*“They can receive up to like, I believe, a 28 day supply of medications if they show up with their prescription bottle or a prescription from the VA”.*

*(Staff 102)*


Other examples of disaster preparedness included tabletop exercises with HBPC teams, where teams run through potential scenarios and how to respond to them. Some Veterans and caregivers stated they did not remember reviewing disaster plans with their VA care providers. These Veterans tended to be those who lived in the area longer and had experienced similar disasters before.

#### 3.1.1. VA Teams Provided High-Quality Care While Managing Their Own Post-Hurricane Challenges

VA staff also had a formal process to call all Veterans on their panel following the hurricane. In some cases where HBPC staff were unable to make these post-storm calls, other staff and leadership filled in. This collaboration proved important, as the first month after Hurricane Ian proved especially challenging, both because of extensive infrastructure damage in the surrounding areas, as well as team members, Veterans, caregivers, and the community processing the impact of such a strong hurricane. One HBPC staff member noted,


*The month after Hurricane Ian came through was very difficult. Not only because we had staff that were affected by the storms as many of our Veterans were, but even if they weren’t affected, there were so many floods and power outages and road outages that we really had trouble getting back to see the patients. It was a big old nightmare is what it was.*

*(Staff 101)*


Examples of VA staff continuing to provide high quality care despite post-hurricane challenges included that staff travelled to check on the Veterans who they could not reach by phone. One staff member shared,


*The infrastructure took a huge hit. So, it was difficult to get in touch with some people. And again, I think what happened was we had some nurses down here who would drive out, just like I did, to check on people. And as we kept trying to get in touch with people, we found out more and more.*

*(Staff 108)*


Staff members noted that some Veterans were hard to contact post-hurricane because they were transferred to non-VA hospitals, had evacuated and had not returned home yet, and because cell towers were down.


*We know who our patients are that are the most needy, and the ones we’re most concerned about because [of] their acuity levels and those are the guys we got out to see as quickly as possible. My one guy that I couldn’t find, who was up in the hospital in Sarasota, he’s the one I had to get out to as soon as I could, because we have hard time telephone communicating with him as it is.*

*(Staff 109)*


Another staff member shared a similar challenge:


*We have people [HBPC Veterans] that end up going to across the state, go up north… You’re trying to track them down, “Where’d you go?!” So, it was really a big scramble of trying to get in touch with people and tryin’ to determine where they were, what their needs were, that whole thing. Again, there’s all kinds of stories. And it’s sad because people’s lives are so uprooted.*

*(Staff 108)*


VA staff also shared the importance of staff having their own preparedness plans ready in the advance of disasters, so they are safe and better prepared for Veterans during these stressful times. The same staff member stated,


*I’m not gonna say put them ahead of yourself, because you have to take care of yourself… but you gotta get out there and see them as quickly as possible. Just if anything, to reassure them that everything’s OK. We’re gonna be there for them. Help them with anything they need that’s within your scope of practice or get them to the resources, social worker, et cetera.*

*(Staff 108)*


#### 3.1.2. VA Made Specific Changes to Improve Work Flow

HBPC care teams shared that they developed a new work flow process in Microsoft Teams to track Veterans closely, as far as who was contacted and to not be duplicative. One staff member shared,


*When we know we’re gonna activate the system, our program assistant built census lists… they’re alphabetized by nurse so we can report. And because it’s in [Microsoft] Teams everybody can—as long as you have computer access, everybody can access it simultaneously and check a box that they’ve completed the call and there’s a comment section. So, we’ve changed that process dramatically.*

*(Staff 101)*


Some members of the HBPC team also created a new emergency preparedness workgroup after Ian to improve disaster preparedness practices. One staff member working on the workgroup noted areas that had room for improvement.


*“We had never seen the likes of anything like this that truly, truly was like, wow, you know, this was the real deal!”.*

*(Staff 107)*


The main task of the workgroup has been to analyze the emergency preparedness plan and coordinate and collaborate with different subject matter experts such as oxygen, pharmacy, and emergency management at the Bay Pines, FL VAMC. Updated recommendations from the workgroup included two weeks of medications, food and supplies, and adequate oxygen tanks for those who need it. A staff member leading the workgroup noted,


*“We have procedures. We have SOPs. We actually made an addendum to change things because of our experience, where there’s workgroups workin’ on now. I mean it’s—we try to learn”.*

*(Staff 108)*


### 3.2. Veterans and Caregivers Felt Highly Supported by VA Care Teams

Veterans, caregivers, and VA staff provided examples of how the HBPC team members supported Veterans after the storm. Locating the Veterans proved to be a critical part of that effort, and then assessing Veterans’ needs and how the VA could assist. When asked how the HBPC team and the VA could have been more supportive after the storm one Veteran explained,


*“I was very satisfied with the care. They cared, you know, they called, they cared. They worried about us, so I don’t know what more they can do for us”.*

*(Veteran 206)*


Another Veteran noted that her HBPC team asked if she wanted to go to a shelter before the hurricane hit, but she declined. She noted,


*“They really care for me and I can get pretty testy at times, but they’re there”.*

*(Veteran 201)*


As power outages were common, one VA staff member shared an example of supporting a Veteran post-Ian in this way,


*I had this elderly gentleman, he’s so nice. But, his power’s out. He has no phone, no way to charge his phone, but he had cell service, so I ended up getting a charger for him. There’s actually an old laptop that he could plug into the wall and then plug the charger in, or if he could keep the battery charged long enough, you know, use that as an extra charger for his phone, and to use it minimally.*

*(Staff 109)*


Staff members shared that resources were hard to come by, including gas and food. One staff member said that many Veterans did not have adequate food supplies following the hurricane.


*We put our Veterans in touch with food pantries but I can’t transport Veterans in my government car. But we have homebound patients and how are they gonna get to the foodbanks?… And then I, you know, couldn’t really go to food pantries asking for five families worth of food so that I could take it to them… I would’ve been willing to do that but there was a little bit of a logistics issue with getting some people food... So, some of our lower socioeconomic, lower people, really could’ve used maybe some food. But other than to let them know where they could go it, that’s all I could really offer. And there was plenty of places, ‘cuz like I said, a lot of aide comes into town.*

*(Staff 106)*


Another Veteran noted that after Ian had passed,


*“They [VA staff] called me from Bay Pines, that’s our hospital, they called me right away and asked how everything was. I told them it was OK, and then she gave me the phone number to call if I needed something”.*

*(Veteran 206)*


### 3.3. Veterans and Caregivers Experienced Significant Challenges Due to the Hurricane, Relying Heavily on Their Social Support Networks Post-Hurricane

Some Veterans and caregivers experienced significant challenges during and after the hurricane, with a few displaced and having to relocate, and others having difficulty accessing needed oxygen and suffering serious medical complications related to the hurricane.

#### 3.3.1. Accessing Needed Healthcare and Processing the Hurricane

Some of the biggest challenges related to Veterans’ health post-Ian included accessing needed oxygen and dealing with depression and anxiety. One staff member shared that a Veteran who was oxygen dependent died the night of the hurricane after the power went out at his home and police and emergency services could not reach him. This VA staff member noted,


*The problems we have especially, and this happened with the last hurricane as well: oxygen. That’s my big thing because these people, a lot of them are oxygen dependent, they use a concentrator, concentrator needs power, they don’t have power for three to four weeks, they don’t have oxygen.*

*(Staff 109)*


Another shared,


*Some patients have had some anxiety related to the storm. Some depression as well. Primarily on that side of things, people losing their home or having to do significant repairs to their home. That adds a lot of stress to them, and or depression to them as well. And that’s been primarily the thing that I’ve noticed overall.*

*(Staff 102)*


Another staff member reflected on visiting a Veteran and caregiver for the first time post-hurricane.


*I saw a Veteran today that I hadn’t seen since before the hurricane. And I was talking to the caregiver… And she said, “I’ve never been so scared in my life” [as during the hurricane]. And she’s talking to me about the fact that they don’t know what they’re gonna do the next time because this Veteran, his dementia is so bad, he just basically lies there.*

*(Staff 108)*


#### 3.3.2. Decisions to Shelter in Place or Evacuate, and for Some, Displacement

Some Veterans chose to shelter in place, whether they lived alone and or with others. One MFH caregiver packed up her two Veterans and rented a home in a different state and treated the experience like a vacation. She checked in with neighbors to determine when it was safe to return home.


*“I always call the neighbors and ask….is the electricity back on…How’s the house look, is it still standing?”.*

*(Caregiver 305)*


The destruction caused by Hurricane Ian led to many Veterans leaving their homes temporarily and some permanently. One Veteran moved in with his home health aide after the hurricane. A staff member shared that one Veteran lost his home in the hurricane, moved to California for several months, and then back to Florida, where he stayed with friends while his new home was built. Another Veteran who lived in a three story home lost his business, a hair salon, as well as two cars, and his health went downhill as well.


*He lived on the beach, and he had a business on the lower level. He lived in like a, it is actually three stories, and he lived on the second story… and he was 77… [After Ian] The cars are gone, the hair salon is gone… also, his condition has deteriorated since then, you know, I don’t know if it’s stress from this [the hurricane] played a part in it. He also did have some cancer that he chose not, he had been getting treatment and then he decided that he just didn’t want to do the treatment anymore, but did the depression and the anxiety and the stress of everything [with the hurricane], did that play a part in his decision? You know, I don’t, I don’t know, so there, so there have been stories like that.*

*(Staff 110)*


Hurricane Ian also displaced a Veteran and his wife, who served as his primary caregiver. During and after Ian, the Veteran, who was primarily wheelchair bound, had challenges accessing needed dialysis. An HBPC staff member described talking to the caregiver during this time.


*I spoke multiple times to the spouse of one of our Veterans and she was calling—when I called her, when I finally reached her, they were in sort of the back hallway at one of the local hospitals. And her husband, the Veteran, was on dialysis three days a week. And she kept telling me, “He’s due for dialysis, he’s due for dialysis”. And through some miscommunications she never actually mentioned that fact to the hospital staff. So, it kind of was a strange scenario. But the dialysis clinic had just pretty much closed and told everybody go to the local hospital.*

*(Staff 101)*


This Veteran’s caregiver noted that the situation was highly stressful and that she felt panicked, not only to ensure that her Veteran husband received his dialysis care but also because they lost their home in the hurricane and had to move. She shared,


*I feel like I’m just coming off of a lot of stress of being like this all the time. I don’t feel like that right now, but, of course, getting this place [their new home they were renting] was, you know, the hardest thing here is so many people were affected, not a lot of places to stay. The rent is much higher… And, you know, financially I feel like we’re struggling.*

*(Caregiver 301)*


Another staff member shared,


*There was a Veteran, he lived down close to Fort Myers beach, his entire house—they relocated before the storm to their daughter’s place in Cape Coral and they stayed there through the storm, but their place was demolished… it was damaged severely so they couldn’t go back and live there… And here he is, he can’t go back to his home…So, this Veteran was able to get into a rental somewhat quickly. And unfortunately—and he had a lot of health problems and very, very bad COPD. And I don’t think the hurricane helped that…he had to have oxygen. He had to have a concentrator… Honestly, I tend to think, he probably would’ve lived longer if it hadn’t of been for what happened.*



*Interviewer: So, he’s passed away since then?*



*Yeah. Since then. Which is unfortunate. I mean he had severe respiratory problems. But I don’t think everything that they went through helped and he just started struggling even more.*

*(Staff 108)*


Other staff members also shared information about Veterans who lost their homes completely. One said,


*I had… two [Veterans] that lost their homes completely. Thankfully they had evacuated prior to [Ian] but since been unable, you know to return to their homes. The [hurricane] was September 28, [and it] was well into the end of the year, you know, before they were getting permanent housing… that took a lot of time. It was temporary housing situations for many, and then they were able to go home. But then, like I said, for that cluster that I had, you know, they never were able to return home.*

*(Staff 107)*


#### 3.3.3. Social Support Veterans and Caregivers Received

Veterans and caregivers shared experiences of relying heavily on social support networks post-Ian. One Veteran, who was primarily in a wheelchair, lived alone, and had no familial support, noted how his neighbors helped him post-storm, and how he gave food that would go bad because his power was out to others.


*Once everything went out, that was it… I just gave, you know, gave the food to some different [neighbors], next door… well they had power… had a generator over there… So, I gave him most of the stuff… that he could use and… He brought, whatever food, brought food over… That’s what I ate.*

*(Veteran 203)*


Another Veteran received a generator from a friend of her daughter-in-law who lived in California.


*[These friends] benevolently drove down from northern Florida. A stranger. Perfect stranger and brought, brought a Westinghouse best generator because we didn’t have one, and then they left it, he and his wife left it, and went away, and we never knew who they were, except that my daughter-in-law gave us their name because she knew them somehow… Can you imagine perfect strangers drove down on a hot day, brought this lovely brand new generator, left it in the garage with my son, and took off before they could even, he could even figure out where they lived?!.*

*(Veteran 201)*


Veterans who lived with caregivers fulltime, like MFH Veterans, had an extra level of social support. One MFH caregiver noted how fortunate she and her Veterans felt after Ian. They had decided to go out to eat at Red Lobster because it was open.


*We load up, we get in there, and the entire mall is set up for FEMA and all these people, [for the disaster]. I felt so bad, I’m in there, the restaurant was almost empty and we’re in there eating shrimp. And we sat there, and I mean tears literally welled up in our eyes of like, wow, how blessed we are that we weren’t touched in that manner.*

*(MFH Caregiver 305)*


### 3.4. VA Staff Managed the Attitudes of Veterans Not Taking the Hurricane Seriously

Staff noted that many Veterans needed to take the hurricane more seriously, and, as many had lived in the area for decades and experienced hurricanes before, the attitude often would be that the storm would be manageable and that they would shelter in place. One VA staff member noted the importance of ensuring Veterans have evacuation plans, because of the severity of the storm combined with the frailty of most Veterans they cared for,


*They’re gonna actually carry them [evacuation plans] out and not try to stay. Like all Veterans, the ones I was mentioning to you, whose homes were destroyed, that if they didn’t evacuate, they probably would’ve been killed. These are people with significant mobility issues and what not.*

*(Staff 111)*


Another staff member added,


*“We did the pre-storms [calls] trying to get everyone to, you know, take it seriously. Sometimes they do, sometimes they don’t”.*

*(Staff 109)*


A few VA staff members noted the challenges of encouraging their HBPC Veterans to evacuate, and shared frustrations when they did not listen. Providers noted that while they can strongly suggest that Veterans evacuate, they cannot force them.


*“We’ve had patients that, you know, just didn’t want to evacuate that were down by the beach, that got wiped off the face of the earth”.*

*(Staff 102)*


Another noted,


*“I think we do everything we can for them. But, again, they have the right to make bad choices”.*

*(Staff 108)*


One of the Veterans shared their side of the conversation with a VA staff member this way:


*Well, they [VA Staff member] just asked me if I’m leaving or staying, and I told them I was staying and not leaving… They just said, “Are you sure?” And I said, “Yeah, I’m sure”… I said I’ve been through all the other hurricanes, you know. This one ain’t gonna be no difference. Hopefully, there’s no more [hurricanes] like that one.*

*(Veteran 203)*


Another staff member shared their experience of encouraging a Veteran and caregiver to evacuate, who refused.


*They literally, both of them tried to stay on the island, which was just dumb on their part and we said that and once we told them you need to evacuate. Literally, we’re getting phone calls right before the storm hit [from Veteran and caregiver], saying “You guys [VA staff] gotta come get us!” I was like “We can’t. Nobody can. We told you to leave. We don’t have a choice”.*

*(Staff 102)*


One familial caregiver of a Veteran shared they had planned to “ride it out,” but the morning before Ian hit, they learned it was going to potentially be a category 5-level storm. This caused them to drive from Florida to Indiana to stay with family.


*I got up that morning and the hurricane was gonna be a [Category] 5, and it was heading right for us, and I said, “We’re leaving”. He [the Veteran, the caregiver’s father-in-law] didn’t want to leave, and my daughter didn’t want to leave, but I said, “We’re getting out of here”.*

*(Caregiver 207)*


Another VA staff member also shared her own personal experience of staying with some of her older neighbors during Ian, even against her better judgement, because she did not want them to stay alone.


*We were told that it [the hurricane], we were in the evacuation zone, but the people in the building wouldn’t leave. They would not evacuate, and we really didn’t have anywhere to go. I mean, one lady is 90. The other ones are in their late 70′s and the other man is 82, so, so they said they were staying, so against everything that I know to be right, I stayed with them, so we all stayed together and hunkered down and, and we survived.*

*(Staff 110)*


### 3.5. VA Staff, Veterans, and Caregivers Shared Lessons Learned

The lessons learned from VA staff, Veterans, and caregivers included maintaining VA staff levels at full capacity, keeping oxygen supplies on hand and power sources for them, and encouraging Veterans and caregivers to evacuate if a severe storm were to come again. A VA staff member noted that one outcome based on Ian’s severity is that Veterans and caregivers may evacuate sooner rather than sheltering in place in the future. They said,


*You call up people and they say, “Well, I don’t think it’s gonna be that bad”. Okay. And now, fortunately, because what happened is people are not gonna—because of what happened with Ian, they’re gonna be a little bit more thoughtful about it [evacuating] perhaps.*

*(Staff 108)*


A MFH caregiver shared the importance of always being prepared and having a destination to evacuate to prior to a major storm.


*Hurricane season is June 1st to November 30th and same with rainy season, everything else, always make sure that I have a 30-day supply of medication, I always watch in the news in making sure, the biggest thing is, is being prepared… you wanna make sure that you have undergarments and bed pads and anything that you need you pack. So thank God that I make a lotta lists and I did it by individual person and knew exactly what they needed down to toothbrushes, just like you’d pack up for the summer camp. And that’s exactly how we did it.*

*(Caregiver 305)*


One Veteran had a battery and solar-powered radio they could listen to to monitor the hurricane’s changes. This Veteran shared that the radio was key.


*“I’m glad I had it. And I think every household should have it or something like it”.*

*(Veteran 211)*


A MFH caregiver noted that after Hurricane Irma in 2017, her air conditioning had gone out for several days, and that this made her purchase single air conditioning units for each of her three MFH Veterans.


*“My first priority is the boys [her Veterans]… the AC went. I had to go and buy window units because they had no air conditioners anywhere. So I went and bought three just for them, three window units”.*

*(MFH Caregiver 303)*


A VA staff member summarized their lessons learned as,


*“Overall, we’ve done well, but I also realize we’re dealing with nature in a very devastating way and we’re also dealing with people who—you can tell people what they should do, they’re gonna do what they wanna do”.*

*(Staff 108)*


## 4. Discussion

The results from this study show that despite the severity of Hurricane Ian, the older and physically vulnerable Veterans we interviewed felt highly supported by the VA HBPC and MFH care teams in the Lee Country, FL area. Essentially, Veterans receiving care from programs like HBPC and MFH receive an additional layer of protection compared to older adults not in such programs, as Veterans have care team members working to keep them safe, ensure they have preparedness plans in place, and have access to the care and medical supplies they need. The proactive nature of the VA HBPC staff to create a new workgroup to improve future disaster preparedness and response and tracking systems to call Veterans before and after a disaster are examples that other institutions and groups that care for older adults who are primarily homebound can model. This is critical, as past research has shown time and again how older adults with complicated health histories are particularly vulnerable to hurricanes [27,28]. One study conducted in Puerto Rico following Hurricane Maria interviewed Veterans, their caregivers, and VA staff in home-based long-term care settings, regarding how social determinants of health influenced recovery from the hurricane [28]. Applying the social ecological model of health, many interconnected social determinants played a role, including the strength of social networks, VA and local policies and support, the receipt of needed aide, and governmental and non-governmental partnerships. In comparison, individuals in our study also relied heavily on support from their VA care teams as well as their social networks. Further, participants in our study shared many communication challenges post-Ian. Another previous study surveyed homeless and non-homeless Veterans’ communication preferences related to impending extreme disasters, finding both groups preferred phone, TV, text, e-mail, and radio [29]. In our study, healthcare teams primarily relied on phoning before and after disasters, but it is important to note that other modes may also be useful in the future. Such multi-layered care coordination, attention, and strategic communication are especially critical for older adults that are not under the care of highly trained interdisciplinary teams like VA HBPC. 

VA care teams must balance their concerns for Veterans’ and their caregivers’ safety with the fact that Veterans and caregivers have the agency and self-determination to make their own decisions about being adequately prepared before major disasters like Hurricane Ian, and whether they will shelter in place or evacuate. As reported here, a lack of preparation can manifest as not having adequate supplies of oxygen, lacking necessary dialysis treatment, and sustaining injuries related to sheltering in place. A recent publication studying the critical healthcare needs of older adults following Hurricane Ian in Florida identified similar challenges, especially related to the need for oxygen and dialysis and other critical healthcare [30]. They shared that many older adults died either directly, indirectly, or unrelated to Ian, and suggested increased mobile dialysis units and locations for generators to power life-support facilities be available during future disasters. 

Our study also found that VA staff encountered many instances of Veterans or caregivers not believing the hurricane would be as serious as it turned out to be. Managing the attitudes of individuals who may have lived in hurricane-prone areas for sometimes decades is a major challenge. Considering Ian’s severity and the damage it caused, there is some evidence from our study that when the next major disaster is forecasted, those older Veterans and their caregivers who sheltered in place during Ian may evacuate in the wake of future disasters. As climate change leads to more frequent and severe hazard events, this will be a reality some will have to face [31].

Veterans’ and caregivers’ reliance on the social support of their own networks proved to be critical post-Ian, especially for those who were displaced and live alone. An advantage for Veterans receiving care from HBPC and MFH programs is that Veterans live in their own home or a home-like setting with a caregiver. With that advantage comes with the fact that they are not in a setting like an assisted living facility or a nursing home, where there are several staff members who may be able to readily support them. Much research on disasters and older populations have shown their reliance on social capital in times of crisis [28,32,33,34,35,36,37]. In turn, increases in social isolation and loneliness amongst older adults have been studied extensively in recent years as negatively effecting overall health after disasters [38,39]. The strengthening of social infrastructure is something that care teams like home health agencies and local and regional policy makers must consider when ensuring older, vulnerable populations are safe and cared for during and after future disasters.

### 4.1. Implications

The lessons learned from participants in our study should be shared with those in similar areas of the US and beyond, who might benefit from applying similar proactive changes related to disaster preparedness to better support older adults and their caregivers pre- and post-major disasters. This is especially needed, as recent US census data showed the southern US, where many weather-related disasters occur, accounted for 87% of population growth in 2023 [40]. Other recent research showed an increase in migration to areas of the US prone to wildfires as well as large cities with hot summers [41]. Ensuring adequate supplies of medications and food, as well as access to needed healthcare for older adults and individuals with disabilities or limited mobility during and after catastrophic situations, are key. In equal measure, cultivating necessary social support—for older adults, their caregivers, and healthcare teams—is imperative following such events. One recent study focused on the mental health toll Hurricanes Irma and Harvey took on home-based providers, much like those in our study [42]. Future studies should consider the importance of supporting healthcare staff post-major disasters, especially as in-home care increases. Further, there may be opportunities for those in caregiving roles, whether informally like familial caregivers or informally like MFH caregivers, to create a disaster preparedness network to share resources as well as peer support, which might ease healthcare staff load and reduce social isolation. Such networks might be able to provide resources to in-need older adults post-disasters, and remove some of the burden and expectations from healthcare staff to provide for them.

### 4.2. Limitations

The sample for our study, while purposive of Veterans, their caregivers, and healthcare teams, was also convenience and snowball sampling. Thus, we may have missed potential participants who could have provided us with other interesting perspectives. Other limitations include that this study is specific to experiences of older Veterans, their caregivers, and healthcare teams before, during, and after a hurricane, and that all findings may not be applicable to other extreme hazard events, like wildfires, flooding, and extreme heat. With hurricanes, there is often some lead time to prepare and/or evacuate, while flooding and wildfires may occur with more spontaneity.

## 5. Conclusions

Caring for primarily homebound, older adults before, during, and after major disasters like Hurricane Ian is a complicated and multi-faceted task. Efforts to continue to improve care coordination are critical, as is ensuring communication plans are in place, especially in the event that normal communication means are inoperable. Entities like VA HBPC teams who are experienced and have made proactive changes to improve care plans for their older, homebound Veterans in the event of disasters are well posed to share their lessons learned with other groups like home health agencies that are not part of large systems like the VA. Interviewing Veterans, their caregivers, and care teams in-person proved consequential and irreplaceable to obtain their trust and in turn do justice in sharing their stories. This matters, because, as climate change causes more severe and frequent natural hazard events, discussing disaster preparedness, expanding training, and increasing community capacity will be key to better supporting older adults, caregivers, and healthcare teams.

## Figures and Tables

**Figure 1 geriatrics-09-00010-f001:**
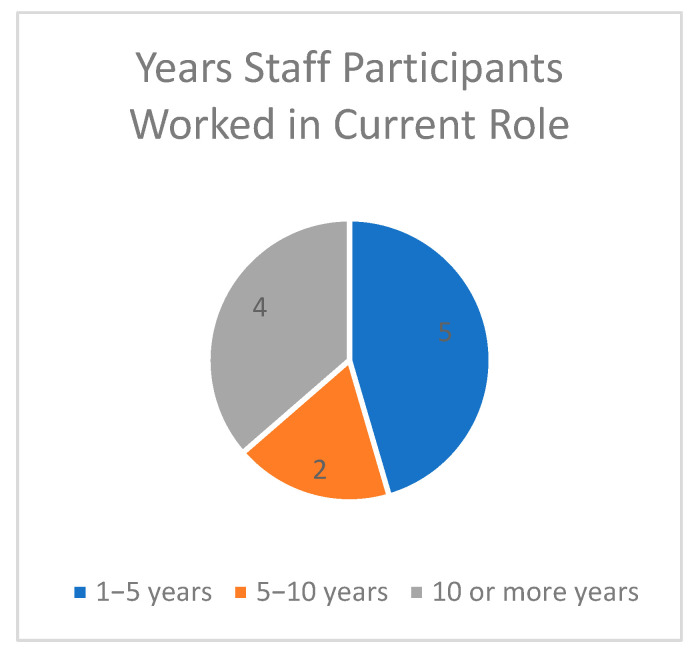
Span of years staff participants worked in their current role.

**Figure 2 geriatrics-09-00010-f002:**
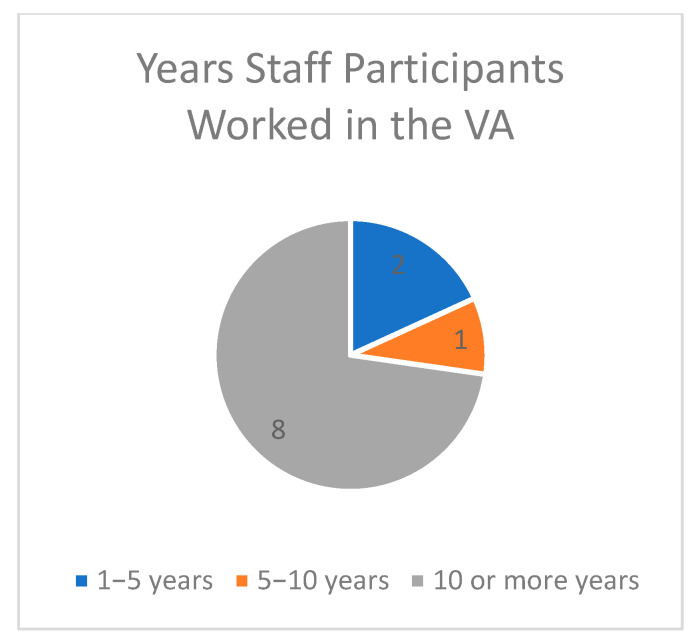
Span of years staff participants worked in the VA.

**Figure 3 geriatrics-09-00010-f003:**
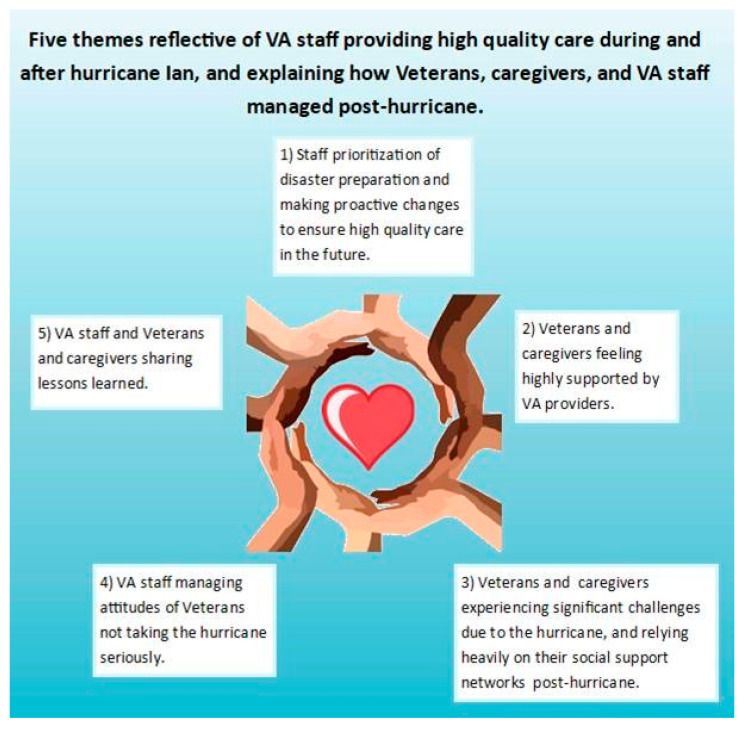
Five themes illustrating how VA staff provided high-quality care during and after hurricane Ian, and explaining how Veterans, caregivers, and VA staff managed post-hurricane.

**Table 1 geriatrics-09-00010-t001:** Veteran Characteristics, N = 8.

Veteran Characteristics		Number
Self-Identified Gender	Male	6
	Female	2
Self-Identified Race	White	7
	Multiracial	1
Hispanic	Hispanic	1
	Non-Hispanic	6
	Declined	1
Age Range	70–85	3
	86–95	4
	Declined	1
Education	High School	4
	Some College	2
	Bachelors	2

**Table 2 geriatrics-09-00010-t002:** Caregiver Characteristics, N = 5.

Caregiver Characteristics		Number (N = 5)
Sex	Male	1
	Female	4
Race	White	4
	Black/African American	1
Hispanic	Hispanic	0
	Non-Hispanic	5
	Declined	0
Age Range	50–60	1
	61–70	2
	Declined	2
Education	High School	1
	Some College	1
	PhD	1
	Declined	2

## Data Availability

As this study is a quality improvement project for the US Department of Veterans Affairs, the data sets cannot be shared publicly.

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
