# Peer review of "Experiences of Veterans, Caregivers, and VA Home-Based Care Providers before, during, and Post-Hurricane Ian"

_geriatrics, 2024, doi:10.3390/geriatrics9010010_

Round 1
Reviewer 1 Report
Comments and Suggestions for Authors
Review of “Experiences of Veterans, Caregivers, and VA Home-Based Care Providers Post-Hurricane Ian” for Geriatrics
This article offers an amazingly detailed examination of the experiences of medically homebound Veterans, their caregivers, and providers with the VA home-based primary care program during and after Hurricane Ian. This insightful look covers the period before, during, and after Hurricane Ian, so I’m not sure if the title needs to be revised to not sell the article short. However, this work was extremely well-written, the methods were thorough and sound, and the results gave a valuable perspective on the challenges faced by both VA staff and Veteran caregivers during disasters. It shines a favorable light on the extensive network, both formally through the Department of Veterans Affairs, and informally, needed to support homebound individuals during disasters, and is an incredible contribution.
One major theme is the time and labor-intensive nature of the VA HBPC staff in reaching out to every single Veteran in its roster by phone. This level of support is undoubtedly valuable to both Veterans and their caregivers who are living independently, without the staffing levels that would be available in a facility, and it would be interesting to consider whether that level of time and labor is sustainable, as disasters become more frequent. The perspective of caregivers, who are often isolated but all facing parallel challenges in disasters, is also interesting. This may be beyond the scope, but it would be interesting to consider whether connecting caregivers in a disaster network, for peer support, might be a good way to leverage resources, ease VA staff burden, and reduce their isolation.
Given the heart-wrenching stories presented in the findings, many of which are beyond the ability of health care institutions like VA or even community first responders to address, the implications, limitations, and conclusion could be more substantial. The current implications section is only 2 sentences, which seems anticlimactic after 7 pages of stories of struggle and difficulty. Is it possible that nothing can be done to address issues of disaster food insecurity for homebound Veterans, many of whom are understandably needy? Would it be possible for health care professionals to coordinate care more efficiently, to reduce stress on their caregivers? Or is this simply the best that our nation’s health care system can do for our most vulnerable? (And if that is the case, it is also an “implication”.)
The limitations and conclusion need to be more thoroughly fleshed out. The fact that it was a convenience sample does not only mean that “interesting perspectives” may be missed—all qualitative research has that limitation because not everyone in that category can be interviewed. Are there also limitations on generalizability to other types of natural disasters, or even other hurricane-impacted regions in the U.S.? The fact that the research team conducted in person interviews is a major strength of the study. Both strengths and limitations should be enumerated in this section. Also, the conclusion being only one sentence suggests that it was tacked on as an afterthought. Surely the authors can leave us with a bit more.
Finally, these may be copy-editing issues but 2 things:
1) There are many spelling errors in the quotes. Since the rest of the manuscript is impeccable, this error probably lies with the transcriptionist, not the authors. (The respondent says “their” instead of “they’re (page 6) or “aid” instead of “aide”, but these nuances were missed.) Nonetheless, these quotes should be cleaned prior to publication. 2) The font size is inconsistent throughout the manuscript.
Comments on the Quality of English LanguageThe manuscript is very well written.
Author Response
Review of “Experiences of Veterans, Caregivers, and VA Home-Based Care Providers Post-Hurricane Ian” for Geriatrics
1. This article offers an amazingly detailed examination of the experiences of medically homebound Veterans, their caregivers, and providers with the VA home-based primary care program during and after Hurricane Ian. This insightful look covers the period before, during, and after Hurricane Ian, so I’m not sure if the title needs to be revised to not sell the article short.
Author Response:
1. Thank you. We have revised the title as suggested.
2. However, this work was extremely well-written, the methods were thorough and sound, and the results gave a valuable perspective on the challenges faced by both VA staff and Veteran caregivers during disasters. It shines a favorable light on the extensive network, both formally through the Department of Veterans Affairs, and informally, needed to support homebound individuals during disasters, and is an incredible contribution. One major theme is the time and labor-intensive nature of the VA HBPC staff in reaching out to every single Veteran in its roster by phone. This level of support is undoubtedly valuable to both Veterans and their caregivers who are living independently, without the staffing levels that would be available in a facility, and it would be interesting to consider whether that level of time and labor is sustainable, as disasters become more frequent.
Author Response:
2. Thank you for this point. We have added in the conclusion section that other groups like home health agencies that are not part of larger healthcare systems can and should learn from VA HPC through HBPC sharing lessons learned.
3. The perspective of caregivers, who are often isolated but all facing parallel challenges in disasters, is also interesting. This may be beyond the scope, but it would be interesting to consider whether connecting caregivers in a disaster network, for peer support, might be a good way to leverage resources, ease VA staff burden, and reduce their isolation.
Author Response:
3. Thank you for this thoughtful comment. We have added a sentence to the end of the implications section to note the possibilities of creating these networks.
4. Given the heart-wrenching stories presented in the findings, many of which are beyond the ability of health care institutions like VA or even community first responders to address, the implications, limitations, and conclusion could be more substantial. The current implications section is only 2 sentences, which seems anticlimactic after 7 pages of stories of struggle and difficulty. Is it possible that nothing can be done to address issues of disaster food insecurity for homebound Veterans, many of whom are understandably needy? Would it be possible for health care professionals to coordinate care more efficiently, to reduce stress on their caregivers? Or is this simply the best that our nation’s health care system can do for our most vulnerable? (And if that is the case, it is also an “implication”.)
Author Response:
4. Thank you for these comments. Based on Reviewer 2’s comments as well, we have extended the implications section and the conclusion to address these comments and strengthen both of these sections.
5. The limitations and conclusion need to be more thoroughly fleshed out. The fact that it was a convenience sample does not only mean that “interesting perspectives” may be missed—all qualitative research has that limitation because not everyone in that category can be interviewed. Are there also limitations on generalizability to other types of natural disasters, or even other hurricane-impacted regions in the U.S.? The fact that the research team conducted in person interviews is a major strength of the study. Both strengths and limitations should be enumerated in this section. Also, the conclusion being only one sentence suggests that it was tacked on as an afterthought. Surely the authors can leave us with a bit more.
Author Response:
5. Thank you for this comment. We have expanded the conclusions section as noted in the comment above and we also have added content to the limitations section. We feel the paper is much improved based on these thoughtful comments.
6. Finally, these may be copy-editing issues but 2 things:
1) There are many spelling errors in the quotes. Since the rest of the manuscript is impeccable, this error probably lies with the transcriptionist, not the authors. (The respondent says “their” instead of “they’re (page 6) or “aid” instead of “aide”, but these nuances were missed.) Nonetheless, these quotes should be cleaned prior to publication. 2) The font size is inconsistent throughout the manuscript.
Author Response:
6. Thank you. We have corrected the their vs they're in the quotes. When participants used slang like "gonna" we kept those. We have corrected font sizes and will work with the journal to format the manuscript correctly.
Reviewer 2 Report
Comments and Suggestions for Authors
The following are my comments for improvement of this paper:
1. The related works are very briefly reviewed in just one paragraph. Please elaborate on the same.
2. Section 2.3 Data Analysis is missing details related to the methodology. Please include a flowchart to show the step-by-step process that was followed for data analysis.
3. The Implications section presents just a couple of implications of the findings, which seems inadequate. Furthermore, one of the implications is – “Ensuring adequate supplies of medications and access to needed healthcare for older adults during catastrophic situations are key, just as is having necessary social support—for older adults, their caregivers, and healthcare teams—is following such events” – This is something which seems applicable for not just older adults but for other diversity groups (such as disabled or handicapped individuals) who experience a catastrophic event. Consider updating this implication to make it more specific to older adults.
4. In the context of older adults and older veterans, the authors state – “Many of these individuals are primarily homebound and often live alone”. A proper discussion of the concerns that affect the elderly – for instance, loneliness, issues with performing daily routine activities, cognitive impairment, etc. is missing. Briefly discuss some of these issues by reviewing some recent works in this area such as https://doi.org/10.1007/978-3-030-55307-4_45 and https://doi.org/10.1109/SENSORS47125.2020.9278630
5. The Conclusions section just contains 1 sentence. The conclusion should be at least one paragraph.
6. A comparison with prior works is missing: Please include a comparative study (qualitative and quantitative) with prior works in this field to highlight the novelty of this work.
7. The presentation of the results in Table 1 could improve if the same is presented using a pie chart or bar graph instead of a table(s).
Author Response
Response to Reviewer 2’s comments
- The related works are very briefly reviewed in just one paragraph. Please elaborate on the same.
Author Response:
1. We have added more background on past studies on homebound, medically complex older adults and navigating major disasters in the introduction as requested. This includes additional citations.
- Section 2.3 Data Analysis is missing details related to the methodology. Please include a flowchart to show the step-by-step process that was followed for data analysis.
Author Response:
2. We greatly appreciate the reviewer’s comment, however, we feel section 2.3 is written very clearly outlining the steps we used to analyze our interview data. We applied a very rigorous approach as we explain in section 2.3. As this is not a methods paper, we do not feel that adding a flowchart illustrating the data analysis adds to the main purpose of the paper. - The Implications section presents just a couple of implications of the findings, which seems inadequate. Furthermore, one of the implications is – “Ensuring adequate supplies of medications and access to needed healthcare for older adults during catastrophic situations are key, just as is having necessary social support—for older adults, their caregivers, and healthcare teams—is following such events” – This is something which seems applicable for not just older adults but for other diversity groups (such as disabled or handicapped individuals) who experience a catastrophic event. Consider updating this implication to make it more specific to older adults.
Author Response
3. Thank you for these thoughtful comments. We have added new citations in the implications section highlighting recent census data showing that the Southern US, where many US weather-related disasters occur, had 87% of national population growth in 2023. We also added explanations and citation about recent research showing that more people are moving to wildfire prone areas as well as cities which have hotter summers. These citations add support to why the implications of our study matter. To address the second comment, we have added clarification that the lessons learned from our study can be extended to populations living with disabilities or limited mobilities as well. We appreciate the reviewer noting the need for this change. - In the context of older adults and older veterans, the authors state – “Many of these individuals are primarily homebound and often live alone”. A proper discussion of the concerns that affect the elderly – for instance, loneliness, issues with performing daily routine activities, cognitive impairment, etc. is missing. Briefly discuss some of these issues by reviewing some recent works in this area such as https://doi.org/10.1007/978-3-030-55307-4_45 and https://doi.org/10.1109/SENSORS47125.2020.9278630
Author Response:
4. Thank you for these comments. We have added additional explanation in the introduction on the topics the reviewer suggested as well as additional citations to note impact of being homebound combined with loneliness, cognitive impairment, difficulty with activities of daily living, lack of social support and other factors. - The Conclusions section just contains 1 sentence. The conclusion should be at least one paragraph.
Author Response:
5. We agree with the reviewer and have improved the conclusions section of the manuscript. - A comparison with prior works is missing: Please include a comparative study (qualitative and quantitative) with prior works in this field to highlight the novelty of this work.
Author Response:
6. We appreciate this comment. We have expanded on a comparative qualitative manuscript that studied VA home-based long-term care settings post Hurricane Maria in the discussion section. We had cited this manuscript in the discussion and to strengthen this reviewer’s request, we have explained and compared the findings. We also added a description of findings of a quantitative, survey study identifying communication preferences of homeless and non-homeless Veterans to receive information about impending extreme disasters to communication challenges in our study. - The presentation of the results in Table 1 could improve if the same is presented using a pie chart or bar graph instead of a table(s).
Author Response:
7. Thank you for this suggestion. We have created two figures/pie charts to represent Table 1 in place of the Table.
Reviewer 3 Report
Comments and Suggestions for Authors
I think this an excellent, original paper, methodologically sound, thorough in its approach and clearly and convincingly argued. It provides valuable advice to those responsible for the care of older people in severe hurricanes and other hard situations. The authors correctly identify certain limitations to their study which future research can seek to fulfil. This is a normal characteristic of original research which opens up opportunities for further research. There a very few minor errors in the writing, which are hard to avoid and easily corrected. With these very minor corrections I believe that the paper merits publication.
Comments on the Quality of English LanguageI think this an excellent, original paper, methodologically sound, thorough in its approach and clearly and convincingly argued. It provides valuable advice to those responsible for the care of older people in severe hurricanes and other hard situations. The authors correctly identify certain limitations to their study which future research can seek to fulfil. This is a normal characteristic of original research which opens up opportunities for further research. There a very few minor errors in the writing, which are hard to avoid and easily corrected. With these very minor corrections I believe that the paper merits publication.
Author Response
Review comment:
I think this an excellent, original paper, methodologically sound, thorough in its approach and clearly and convincingly argued. It provides valuable advice to those responsible for the care of older people in severe hurricanes and other hard situations. The authors correctly identify certain limitations to their study which future research can seek to fulfil. This is a normal characteristic of original research which opens up opportunities for further research. There a very few minor errors in the writing, which are hard to avoid and easily corrected. With these very minor corrections I believe that the paper merits publication.
Author Response:
- Thank you for the kind comments. We have corrected minor errors throughout as requested.
Round 2
Reviewer 2 Report
Comments and Suggestions for Authors
The authors have revised their paper as per all my comments and feedback. I do not have any additional comments at this point. I recommend the publication of this paper in its current form.